# Beyond the Growth: A Registry-Based Analysis of Global Imbalances in Artificial Intelligence Clinical Trials

**DOI:** 10.3390/healthcare13162018

**Published:** 2025-08-16

**Authors:** Chan-Young Kwon

**Affiliations:** 1Department of Oriental Neuropsychiatry, Dong-Eui University College of Korean Medicine, 52–57, Yangjeong-ro, Busanjin-gu, Busan 47227, Republic of Korea; beanalogue@naver.com; Tel.: +82-51-850-8808; 2Anti-Aging Research Center, Dong-Eui University, 52–57, Yangjeong-ro, Busanjin-gu, Busan 47227, Republic of Korea

**Keywords:** artificial intelligence, machine learning, digital therapeutics, clinical trials, WHO ICTRP

## Abstract

**Background/Objectives**: While the integration of artificial intelligence (AI) into clinical research is rapidly accelerating, a comprehensive analysis of the global AI clinical trial landscape has been limited. This study presents the first systematic characterization of AI-related clinical trials registered in the World Health Organization (WHO) International Clinical Trials Registry Platform (ICTRP). It aims to map global trends, identify patterns of concentration, and analyze the structure of international collaboration. **Methods**: A search of the WHO ICTRP was conducted on 20 June 2025. Following a two-stage screening process, the dataset was analyzed for temporal trends, geographic distribution, disease and technology categories, and international collaboration patterns using descriptive statistics and network analysis. **Results**: We identified 596 AI clinical trials across 62 countries, with registrations growing exponentially since 2020. The landscape is defined by extreme geographic concentration, with China accounting for the largest share of trial participations (35.6%), followed by the USA (8.5%). Research is thematically concentrated in Gastroenterology (22.8%) and Oncology (20.1%), with Diagnostic Support (45.6%) being the most common technology application. Formal international collaboration is critically low, with only 8.7% of trials involving multiple countries, revealing a fragmented collaboration landscape. **Conclusions**: The global AI clinical trial landscape is characterized by rapid but deeply imbalanced growth. This concentration and minimal international collaboration undermine global health equity and the generalizability of AI technologies. Our findings underscore the urgent need for a fundamental shift toward more inclusive, transparent, and collaborative research models to ensure the benefits of AI are realized equitably for all of humanity.

## 1. Introduction

The integration of artificial intelligence (AI) into clinical medicine has entered a pivotal phase, characterized by exponential growth in research activity and increasing recognition of AI’s transformative potential across healthcare delivery [1,2]. While early AI applications in medicine focused primarily on retrospective analyses and controlled laboratory settings [3], the field has rapidly evolved toward prospective clinical evaluation and real-world implementation. This evolution reflects both the maturation of AI technologies and growing confidence in their clinical utility, particularly in diagnostic applications where AI systems have demonstrated performance comparable to or exceeding human experts [4,5,6,7].

However, the translation of AI innovations from research prototypes to clinical practice reveals a complex landscape of geographic disparities, methodological challenges, and equity concerns [8,9]. Recent comprehensive analyses have illuminated critical patterns in AI health research that extend beyond simple technological advancement. For instance, recent data indicates this disparity extends to the development of frontier technologies, with U.S.-based institutions significantly outpacing other regions in notable AI model development [10]. This concentration of cutting-edge model development suggests that the translation of AI innovations is geographically skewed, potentially influencing which populations benefit most from technological advances. This geographic stratification in research quality and impact suggests fundamental disparities in the AI health research ecosystem that may influence how AI technologies are developed, validated, and ultimately deployed in clinical practice.

The clinical evaluation of AI technologies has followed similar geographic patterns. A recent scoping review of randomized controlled trials (RCTs) evaluating AI in clinical practice found that while 81% of trials reported favorable outcomes, significant concerns persist regarding generalizability, publication bias, and the predominance of single-center studies [8]. Geographic analysis revealed that the United States and China dominated clinical AI trials, with China leading in gastroenterology applications while the United States maintained a more diverse specialty distribution [8]. These findings underscore a critical implementation gap, as despite promising technical performance, most AI systems remain geographically concentrated in their development and testing, potentially limiting their global applicability [1].

The implications of this geographic concentration extend beyond research productivity to fundamental questions of health equity and technology access. Recent evidence suggests that AI research concentration in specific regions may create biased data foundations that distort inferences and potentially lead to biased medical care [11,12]. The underrepresentation of certain populations in AI training datasets has already been shown to lead to diagnostic disparities, with AI algorithms performing less accurately for underrepresented ethnic groups [13]. As AI clinical research has accelerated exponentially, particularly since 2020 [8], these equity concerns have become increasingly urgent.

The World Health Organization (WHO) International Clinical Trials Registry Platform (ICTRP) represents the most comprehensive repository of clinical trials data globally, aggregating information from over 20 national and regional registries (url: https://trialsearch.who.int/) [14,15]. This platform provides an opportunity to conduct systematic analysis of AI clinical research at a global scale, transcending the limitations of single-registry studies and offering insights into patterns of research activity across diverse healthcare systems and regulatory environments.

This study offers a comprehensive analysis of AI-related clinical trials in the WHO ICTRP, aiming to map global trends, identify disease-technology patterns, and assess implications for equity, regulation, and research priorities.

## 2. Materials and Methods

### 2.1. Data Sources and Search Strategy

A systematic search was conducted in the WHO ICTRP Advanced Search interface on 20 June 2025. All clinical trial protocols registered by the search date were considered. The search strategy employed Boolean operators in the Intervention field using the following terms: “Artificial Intelligence” OR “AI” OR “Machine Learning” OR “Deep Learning”. Data were downloaded in XML format for subsequent analysis. Text preprocessing involved HTML tag removal, special character normalization, and whitespace standardization to ensure consistent data formatting across all trial records.

### 2.2. Study Selection and Screening

The researchers screened the initial dataset to identify AI-related clinical trials. Each potentially relevant trial was assessed against predefined criteria assessing the centrality of the AI component in the study intervention or diagnostic approach. Only studies using AI as a core intervention or diagnostic tool were included. Studies without a clear AI/machine learning/deep learning component, studies using AI only for post hoc data analysis, studies using existing medical interventions without an AI component, studies mentioning AI only peripherally, and studies where the AI intervention was ambiguous were excluded. The screening process was conducted systematically by a single investigator (C.-Y.K.) using standardized evaluation forms. To ensure consistency, all borderline cases were re-evaluated against the inclusion criteria, and detailed rationales for inclusion/exclusion decisions were documented. A random sample of 50 trials (7.7%) was re-screened after a 2-week interval to assess intra-rater reliability, achieving 100% agreement.

### 2.3. Data Analysis

#### 2.3.1. Geographic Distribution

For geographic distribution analysis, we employed a ‘full count approach’ where each country participating in a multinational trial was counted separately. For example, if a single trial involved collaboration between the United States, Germany, and France, this trial would contribute one count to each of these three countries’ totals, resulting in three total counts from one unique trial. Consequently, the total number of national participations is greater than the unique number of trials, and all reported percentages for geographic distribution are based on this total number of participations.

This contrasts with ‘lead country only’ approaches that would assign the entire trial to only the primary investigator’s country. This methodology was necessary because traditional approaches systematically underrepresent the contributions of collaborating nations, thereby obscuring true patterns of international research participation and cooperation. The full count approach provides a more comprehensive understanding of which countries are actively engaged in AI clinical research, regardless of their role as lead or collaborating institutions. In contrast, disease and technology category analyses were based on unique trial counts to prevent duplication bias in thematic analyses, ensuring that each trial contributed only once to category totals.

#### 2.3.2. Topic Modeling

Disease state-centric analyses investigating key health conditions and technology intervention-centric analyses investigating AI modalities and applications were performed. Topic modeling was conducted using BERTopic with specialized biomedical embeddings (pritamdeka/S-BioBert-snli-multinli-stsb) [16,17]. The BERTopic algorithm employed UMAP for dimensionality reduction (n_neighbors = 5, n_components = 5, min_dist = 0.0, metric = ‘cosine’), HDBSCAN for clustering (min_cluster_size = 5, min_samples = 3, metric = ‘euclidean’), and CountVectorizer for feature extraction (min_df = 2, max_features = 1000, ngram_range = (1, 2)). The number of topics was determined automatically by the algorithm’s data-driven clustering approach (Table 1). The final topic classifications underwent manual review and assignment to predefined disease and technology taxonomies. Topics were refined to ensure clinically meaningful classifications, with outlier topics classified as “Others”.

#### 2.3.3. Network Analysis

International collaboration patterns were analyzed using network analysis methods implemented in NetworkX (version 2.8) [18]. Countries were represented as nodes, with edge weights representing the frequency of collaborative partnerships between country pairs. Network centrality measures and temporal collaboration patterns were calculated to identify key collaborative hubs and assess the evolution of international partnerships over time. Network visualizations used node sizes to represent either total trial participation (full count approach) or collaboration centrality, while node colors indicated temporal patterns of research activity based on average research years.

### 2.4. Statistical Analysis

Descriptive statistics were calculated for temporal trends, geographical distribution, disease categories, and technology applications. Time series analysis was used to analyze growth patterns and inflection points in AI clinical trial enrollment. Disease-technology association patterns were analyzed using cross-tabulation and visualized using heatmap matrices. For visualization clarity, association matrices focused on the most prevalent disease and technology categories, with less frequently represented categories grouped as “Others”. Statistical validation of temporal trends included Mann–Whitney U tests and *t*-tests comparing pre-2020 (2017–2019) versus post-2020 (2020–2024) trial registrations, chi-square tests for temporal distribution patterns, and linear regression analysis to assess structural breaks at the 2020 inflection point. Effect sizes were calculated using Cohen’s d, with statistical significance set at *p* < 0.05. All analyses were performed using Python 3.12 with the following libraries: Pandas for data manipulation, scikit-learn for machine learning utilities, BERTopic for topic modeling, matplotlib and seaborn for visualization, sentence-transformers for text embedding, NetworkX for network analysis of international collaborations, UMAP for dimensionality reduction, HDBSCAN for clustering, and NumPy for numerical computations.

## 3. Results

### 3.1. Study Selection and Characteristics

From an initial dataset of 648 trials, our screening process identified 596 trials (92.0%) as AI-related, while 52 (8.0%) were excluded based on relevance criteria. The final analysis included 596 AI-related trials conducted in 62 countries from 2011 to 2025, representing 668 total country participations when accounting for multinational collaborations, covering 12 major disease categories and seven major technology categories. Enrollment in AI trials has shown an exponential growth pattern, particularly accelerating since 2020. The earliest AI trials were initiated in 2011 (n = 1), with minimal activity until 2017 (cumulative n = 8).

To validate the observed temporal patterns, we conducted statistical analyses comparing pre-2020 (2017–2019) and post-2020 (2020–2024) periods. The mean annual trial registrations increased dramatically from 10.0 trials per year pre-2020 to 96.4 trials per year post-2020, representing a 9.6-fold increase. This difference was statistically significant (Mann–Whitney U test: *p* = 0.018; independent *t*-test: *p* = 0.028) with a large effect size (Cohen’s d = 2.10). Temporal distribution analysis revealed a significant departure from random distribution patterns (χ^2^ = 464.2, *p* < 0.001), with 94.1% of all trials (561/596) registered in the post-2020 period despite representing only 42% of the study timeframe. Structural break analysis confirmed 2020 as a significant inflection point, with the annual registration slope increasing from 7.0 trials/year (2017–2019) to 29.3 trials/year (2020–2024), representing a 4.2-fold acceleration in registration velocity (both slopes *p* < 0.05) (Figure 1).

### 3.2. Disease and Technology Category Distribution

The 596 AI trials were distributed across 12 major disease categories. Gastroenterology had the largest number of trials, with 136 (22.8%), followed by Oncology with 120 (20.1%). Endocrinology and Cardiovascular Medicine each had 60 trials (10.1%). Pulmonary Medicine had 46 trials (7.7%) and Musculoskeletal Medicine had 29 trials (4.9%). Neuroscience had 21 trials (3.5%), Mental Health and Urology each had 20 trials (3.4%). Smaller categories included Infectious Diseases with nine trials (1.5%), Ophthalmology with five trials (0.8%), and Dermatology with five trials (0.8%). The remaining 65 trials (10.9%) were classified as “Other” (Figure 2a). The analysis results in seven major technology categories. Diagnostic support was the most common, with 272 cases (45.6%), followed by digital therapeutics and patient management, which accounted for 92 cases (15.4%) and surgical and procedural assistance (85, 14.3%). Medical imaging analysis accounted for 40 cases (6.7%), drug development and biomarker discovery accounted for 28 cases (4.7%), and predictive modeling accounted for 26 cases (4.4%). Robotic medicine accounted for the smallest share with five cases (0.8%). The remaining 48 cases (8.1%) were classified as “Others” (Figure 2b).

Representative examples within major categories include: for Gastroenterology, trials evaluating AI-assisted colonoscopy detection systems (NCT06656624) and deep learning algorithms for predicting the risk of digestive diseases (ISRCTN74930639); for Oncology, studies of machine learning-based skin cancer screening (TCTR20211101004) and predicting cancer treatment response (NCT05070884); for Diagnostic Support technology, trials testing AI-enhanced radiology interpretation systems (KCT0005051) and automated pathology diagnosis tools (NCT06773832); and for Digital Therapeutics, studies of cognitive behavioral therapy-based chatbot (ChiCTR2100052532) and AI-based digital therapeutic for diabetic microvascular complications (ChiCTR2500103622).

### 3.3. Geographic Distribution and Global Patterns

Geographic analysis, based on the ‘full count’ of participations, revealed significant differences in AI clinical trial activity across countries. China led with 238 participations (35.6%), followed by the United States with 57 (8.5%), and Germany with 42 (6.3%). The United Kingdom participated in 38 (5.7%) trials, while Australia had 30 (4.5%). Taiwan accounted for 19 (2.8%) trials, and Spain for 17 (2.5%). Hong Kong and India each had 15 (2.2%) participations, with South Korea rounding out the top ten with 14 (2.1%) trials (Table 2).

### 3.4. Disease-Technology Association Patterns

The disease-technology association matrix illustrates the distribution of trials across various disease and technology categories. Oncology demonstrated the highest activity in diagnostic support with 65 trials. This was closely followed by Gastroenterology, which showed significant engagement in surgical and procedural support (49 trials) and also in diagnostic support (40 trials). Endocrinology likewise had a substantial number of trials in diagnostic support, totaling 46, with Cardiovascular Medicine also showing a strong presence in this category with 37 trials. Regarding drug development, Oncology again led with 14 trials, marking the highest contribution in this area. In digital therapeutics and patient management, Pulmonology showed considerable activity with 18 trials, while also having 16 trials in diagnostic support. For diagnostic support, Mental Health and Urology each registered 14 trials, whereas Neuroscience had 3 trials. Further examining drug development, beyond Oncology, Gastroenterology contributed seven trials, and Musculoskeletal Medicine, along with Urology, each had one trial. In the domain of medical imaging, Musculoskeletal Medicine was particularly active with 11 trials, followed by Endocrinology with 8 trials, and Neuroscience with 7 trials. Overall, Dermatology, Ophthalmology, and Infectious Diseases remained the least represented across most technology categories (Figure 3).

### 3.5. Country-Disease and -Technology Association Patterns

The country-disease association matrix shows the distribution of trials across the top 10 countries and the top 10 disease categories. China exhibited the highest number of trials in Gastroenterology (60 trials) and Oncology (69 trials), also showing significant activity in Cardiovascular Medicine (24 trials). The United States displayed a notable number of trials in Mental Health (10 trials) and Cardiovascular Medicine (10 trials), while Germany showed relatively higher numbers in Gastroenterology (14 trials) (Figure 4). Subsequently, the analysis of the country-technology association matrix reveals the distribution of trials across the top 10 countries and technology categories. China dominated in Diagnostic Support with 108 trials and also showed substantial activity in Surgical and Procedural Assistance (48 trials) and Digital Therapeutics and Patient Management (19 trials). The United States had the second-highest number of trials in Diagnostic Support (24 trials) and Digital Therapeutics and Patient Management (13 trials). Germany and the United Kingdom both showed a notable number of trials in Diagnostic Support, with 19 and 20, respectively. These matrices collectively highlight the varying focus and activity of different countries in AI clinical trials across diverse disease and technology areas (Figure 5).

### 3.6. International Collaboration in AI Clinical Trials

Of the AI-related clinical trials, 585 were analyzed, excluding 11 trials for which country information was unavailable. Among these, only 51 trials involved international collaboration, while the majority (534 trials) were conducted in a single country. This indicates an overall low global collaboration rate (approx. 8.7% of analyzed trials), suggesting that most AI trials remain nationally confined despite the international nature of AI development.

The network visualization reveals distinct regional clustering patterns. European countries (Germany, UK, Spain, Italy, France, Belgium) form a densely interconnected cluster, while Asian countries (China, Hong Kong, South Korea) show limited cross-border connections despite substantial individual research activity. China emerges as the most prominent country with an overwhelming 238 trials, displaying the largest node and signifying its leading position in terms of the sheer volume of AI clinical trial research. Other significant contributors, such as the United States (57 trials), Germany (42 trials), and the United Kingdom (38 trials), also appear as large nodes. However, despite its remarkably high research output, China showed an exceptionally low collaboration rate (1.3%) with only three multicountry participations. This is reflected in its connectivity within this network, appearing relatively focused or less diffuse compared to the more interconnected European and North American clusters, suggesting a predominantly domestically oriented research effort (Figure 6a, Table 2). A crucial observation is the marked reduction in China’s node size in Figure 6b compared to Figure 6a, directly aligning with its low collaboration rate (1.3%) and limited multicountry participation (3 trials) as presented in Table 2. Conversely, countries such as Germany (21.4% collaboration rate, 27 total partners), the United States (10.5% collaboration rate, 18 total partners), Spain (notably high 64.7% collaboration rate with 47 total partners), France (20% collaboration rate, 14 total partners), and the United Kingdom (18.4% collaboration rate, 30 total partners) exhibit notably larger nodes. Regarding the temporal aspect, countries with yellow-hued nodes, such as Turkey, Czechia, Bulgaria, and Macao, signify that their average collaboration years are more recent (closer to 2023–2024). In contrast, countries with bluer nodes, including Romania, Egypt, Luxembourg, and Portugal, indicate earlier average collaboration years (closer to 2020–2021) (Figure 6b, Table 2).

## 4. Discussion

This study provides a comprehensive analysis of AI clinical trials registered with WHO ICTRP, revealing 596 trials generating 668 total country participations across 62 countries from 2011 to 2025. Our findings document rapid growth, pronounced geographic concentration, and clear patterns in disease and technology applications that have important implications for global health technology development.

The exponential growth trajectory of registered clinical trials, particularly the dramatic acceleration since 2020, serves as an unambiguous indicator that AI is transitioning from a nascent technology of experimental potential to a mature class of interventions undergoing serious clinical validation. The surge to 159 new trials in 2024 alone signifies that the technology’s clinical utility and the research community’s confidence in it have crossed a critical threshold [1]. A recent scoping review of AI randomized controlled trials found rapid growth in the field, with most trials reporting positive outcomes [8]. However, that analysis focused on published RCTs, while our registry-based approach captures the full scope of planned and ongoing clinical validation efforts, providing a more comprehensive view of clinical development activity. Additionally, our registry-based analysis may be free from potential publication bias [15].

However, this impressive growth narrative is complicated by the landscape’s geographic concentration, which stands as one of the most critical and concerning findings of this study. The consolidation of 39.9% of all trials within China, with the United States following at a distant 8.5%, while vast regions like Africa and Latin America remain virtually unrepresented, paints a stark picture of global disparity. A recent analysis of 397,967 AI life science publications found the United States leading in cumulative output. However, China surpassed the United States in annual publications since 2020, with North America and Europe generating higher-quality research appearing in top-ranking outlets and receiving up to 50% more citations than other regions [9]. Our clinical trials data reveal a more pronounced Chinese concentration (35.6% of participants) than observed in general AI research publications, suggesting different dynamics in clinical validation compared to research publications. Importantly, both analyses identify concerning patterns of international collaboration stagnation, with Schmallenbach et al. documenting that international collaborations have plateaued relative to national research efforts, paralleling our finding of critically low multinational trial collaboration (8.7%) [9]. This convergence between publication and clinical trial patterns suggests that geographic concentration and collaboration deficits represent systemic challenges across the entire AI research ecosystem, from basic research through clinical validation, potentially limiting the global applicability and equitable access to AI health technologies. As previously indicated [12,13], these profound geographic disparities have troubling implications for global health equity, threatening to create AI technologies that are validated by and built for only a fraction of the world’s population. Importantly, Countries from Africa and Latin America collectively accounted for less than 5% of trials in our dataset, despite these regions representing substantial portions of the global population and disease burden [19,20]. These findings strongly suggest the need for multiracial and multinational studies to improve AI fairness in health care as well.

The underlying drivers of this imbalance can be further understood through the thematic concentration of research efforts. The intense focus on gastroenterology (22.8%) and oncology (20.1%) as disease categories, and on diagnostic support (45.6%) as a technology application, is not arbitrary. It is a rational response to the current capabilities of AI, where deep learning models excel when applied to large, standardized, and often visual datasets (e.g., endoscopy videos, computerized tomography scans, pathology slides) to perform pattern recognition tasks [21]. While this pragmatic prioritization of technologically tractable problems expedites progress in certain domains, it concurrently engenders a significant selection bias, concentrating innovation on data-replete medical fields at the expense of numerous other diseases characterized by more complex or less-digitized data.

A detailed examination of country-specific portfolios confirms that these macro-level concentrations are underpinned by distinct national strategies. For instance, China’s massive investment in diagnostic and procedural tools for gastroenterology and oncology signals a state-directed, industrial-scale approach to tackling high-burden diseases in data-replete environments [22]. The United States, in contrast, exhibits a more diversified portfolio with notable activity in areas like mental health and digital therapeutics, suggesting a more pluralistic ecosystem responding to different clinical and commercial incentives [22]. These disparate developmental trajectories are not only shaping distinct research priorities but are also fundamentally influencing the architectural and data-centric paradigms of the resultant AI technologies. This divergence exacerbates the risk of technological fragmentation, creating substantial barriers to the interoperability and transferability of these tools across heterogeneous healthcare systems.

The final piece amplifying these challenges is the profound lack of international collaboration. The finding that only 8.7% of trials are conducted internationally is stark, but the network analysis map visualizes the deeply fractured nature of this reality. It reveals a regionalized landscape with a dense, interconnected Euro-North American cluster existing largely separate from a more isolated Asian research sphere. The map powerfully illustrates the difference between research volume and collaborative influence; while China is the largest node by trial count (Figure 6a), its node size shrinks dramatically when measured by network centrality (Figure 6b), with European nations like Spain and Germany emerging as more critical collaborative hubs. This fragmented research approach, arising from the patterns described above, perpetuates the geographic and data biases. The stark contrast between research volume and collaborative engagement highlights how network centrality varies significantly across regions, with implications for global knowledge transfer and technology dissemination. It fundamentally hinders the creation of diverse, globally representative datasets, making the goal of AI fairness a distant prospect [12].

These findings highlight urgent needs for coordinated international action. Policymakers and funding agencies should consider establishing international AI clinical trial consortia to facilitate collaborative research across underrepresented regions. Standardized data sharing protocols and harmonized regulatory frameworks could reduce barriers to multinational studies. Capacity-building programs targeting countries with high disease burdens, but limited AI research infrastructure would help address geographic disparities. Additionally, funding mechanisms that incentivize international collaboration, such as matching grants for multinational partnerships, could increase the currently low 8.7% collaboration rate and promote more equitable global AI health technology development.

Unlike many existing reviews that rely on published literature [8,9,12,22], our registry-based methodology circumvents publication bias by including all planned, ongoing, and completed trials, thus offering a more accurate and unfiltered snapshot of the true development pipeline [15]. Also, our methodological strength lies in the multifaceted approach, which synergistically combines temporal, geographic, and thematic analyses with a sophisticated network analysis of international collaboration to present a holistic and deeply integrated view of the AI clinical research ecosystem. Despite these strengths, the study has several inherent limitations. First, this analysis relies exclusively on WHO ICTRP registration data, which may not capture all AI clinical research activity worldwide [14,15]. Countries with different registration requirements, practices, or timelines may be underrepresented, and some research may be conducted without registration in international databases. Second, the search strategy, while comprehensive, may have missed trials using alternative terminology or registered before standardized AI terminology became widespread. Early AI research may have used different descriptive terms such as “neural networks,” “computer vision,” “automated systems,” “expert systems,” or “pattern recognition,” potentially leading to systematic underrepresentation of historical AI clinical activities. This limitation may particularly affect our analysis of temporal trends in the earlier years of our study period. Third, geographic distribution findings may be influenced by systematic differences in registration practices, timing requirements, and completeness standards across national registry systems within the WHO ICTRP network. Variations in enforcement levels, technological infrastructure, and administrative efficiency between countries may create apparent disparities in research activity that reflect registry system characteristics rather than actual trial conduct. Sensitivity analyses to account for these registry-specific factors were not feasible due to the limited availability of system-level metadata. Fourth, this analysis focused solely on trial registration patterns without examining trial phases, completion rates, outcome reporting, or clinical impact. The data represent research intentions rather than actual execution, progression toward regulatory approval, or therapeutic success. Our approach cannot differentiate between exploratory early-phase studies and confirmatory pivotal trials, nor assess whether registered trials achieved their intended outcomes or contributed to clinical practice changes. Many registered trials may remain incomplete, unpublished, or fail to progress beyond initial phases, limiting our ability to evaluate the true clinical translation of AI research activity. Fifth, the study employed a temporal cutoff of 20 June 2025, which means the 2025 data (n = 79) represent only partial-year registrations. This may affect the interpretation of recent trends and growth patterns. Finally, our topic modeling analysis relied on manual clinical validation rather than quantitative coherence metrics. It did not include comparison with alternative clustering methods, which represents a limitation in methodological validation that future studies should address.

## 5. Conclusions

This first comprehensive analysis of the WHO ICTRP reveals that the global landscape of AI clinical trials is defined by two paradoxical trends: explosive growth and profound imbalance. While the rapid acceleration of trials since 2020 signals the maturation of AI in medicine, this progress is geographically concentrated in a few nations, particularly China, and thematically focused on specific areas like diagnostics and oncology. The alarmingly low rate of international collaboration suggests that most AI health technologies are being developed and validated in isolated ecosystems. This reality poses a significant threat to global health equity, risking the development of biased AI systems that fail to generalize across diverse populations. To realize the full, equitable potential of AI in healthcare, a global paradigm shift is urgently needed—one that fosters international collaboration, promotes data diversity, and prioritizes transparent, inclusive research practices.

Future research should examine mechanisms to reduce geographic concentration and enhance international collaboration in AI clinical trials. Longitudinal tracking of clinical outcomes across diverse populations will be essential for developing truly generalizable AI technologies. Additionally, investigating standardized regulatory frameworks and incentive structures for multinational partnerships could help address the current fragmentation in global AI clinical research ecosystems.

## Figures and Tables

**Figure 1 healthcare-13-02018-f001:**
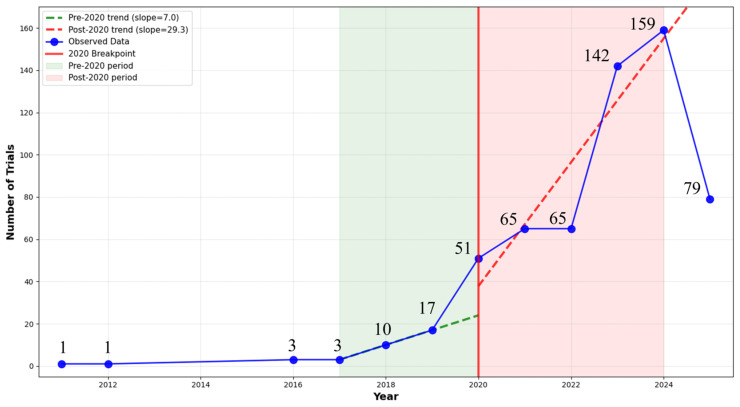
Annual trends of AI clinical trials.

**Figure 2 healthcare-13-02018-f002:**
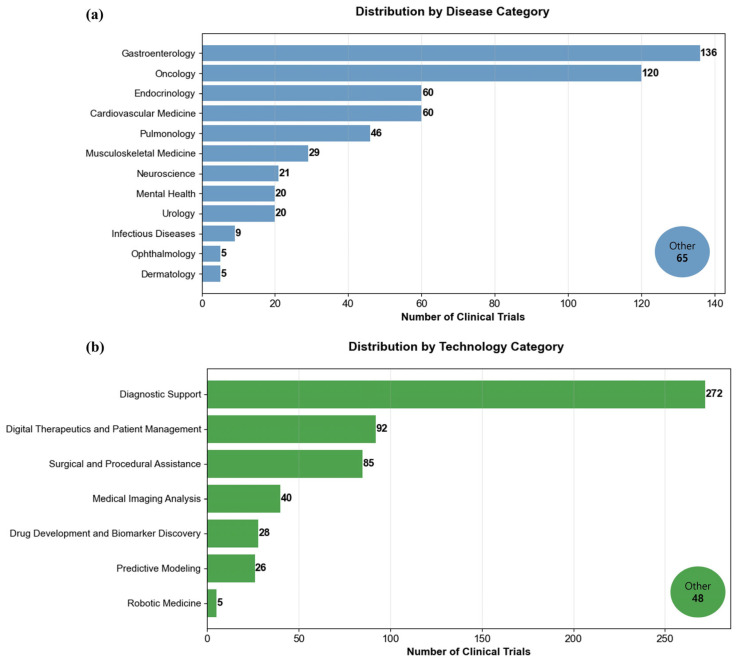
Distribution by disease and technology categories. Note. Horizontal bar chart showing the number of artificial intelligence clinical trials across (**a**) 12 primary disease categories or (**b**) technology applications.

**Figure 3 healthcare-13-02018-f003:**
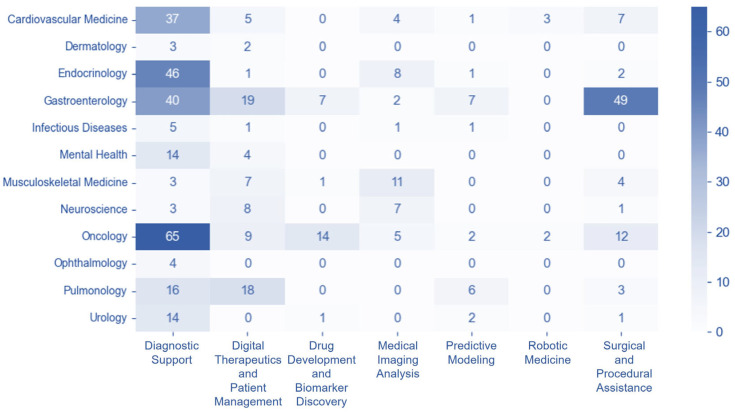
Disease-technology association matrix. Note. Heatmap showing the intersection of disease categories (*y*-axis) with technology categories (*x*-axis). Color intensity represents the number of trials at each intersection, with darker red indicating higher trial concentrations.

**Figure 4 healthcare-13-02018-f004:**
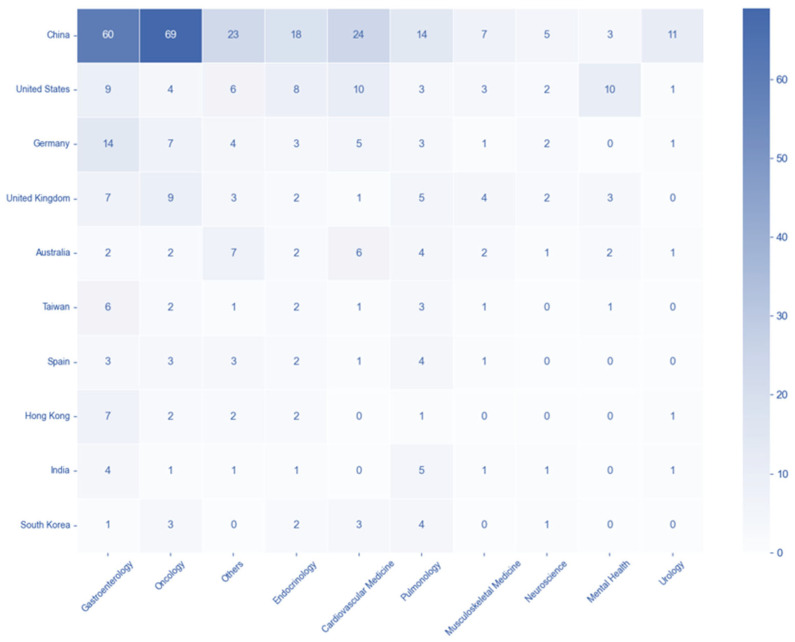
Geographic distribution of AI clinical trials across disease categories. Note. Heatmap displaying the intersection of countries (*y*-axis) and disease categories (*x*-axis). Color intensity represents the number of AI-related clinical trials, with darker blue indicating higher trial concentrations.

**Figure 5 healthcare-13-02018-f005:**
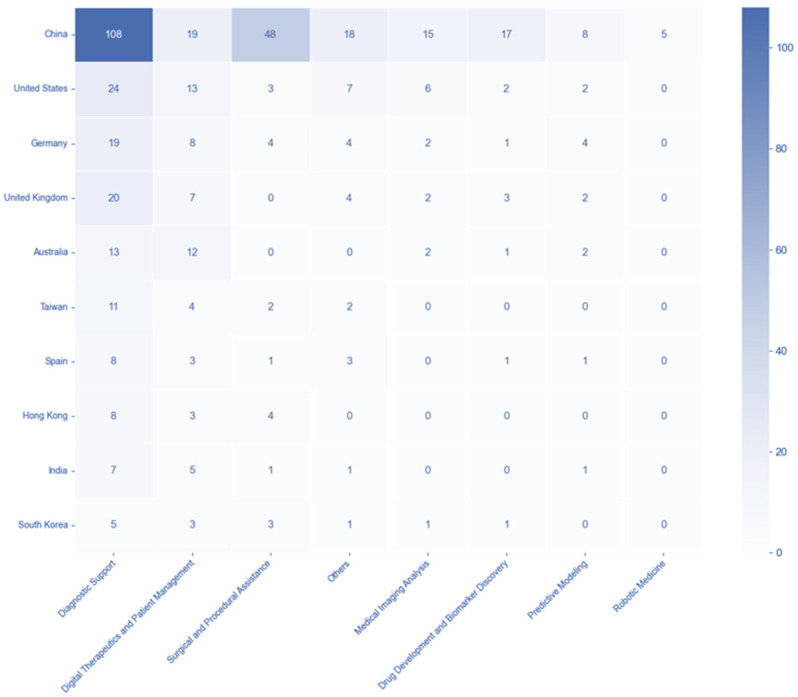
Geographic distribution of AI clinical trials across technology categories. Note. Heatmap displaying the intersection of countries (*y*-axis) and technology categories(*x*-axis). Color intensity represents the number of AI-related clinical trials, with darker blue indicating higher trial concentrations.

**Figure 6 healthcare-13-02018-f006:**
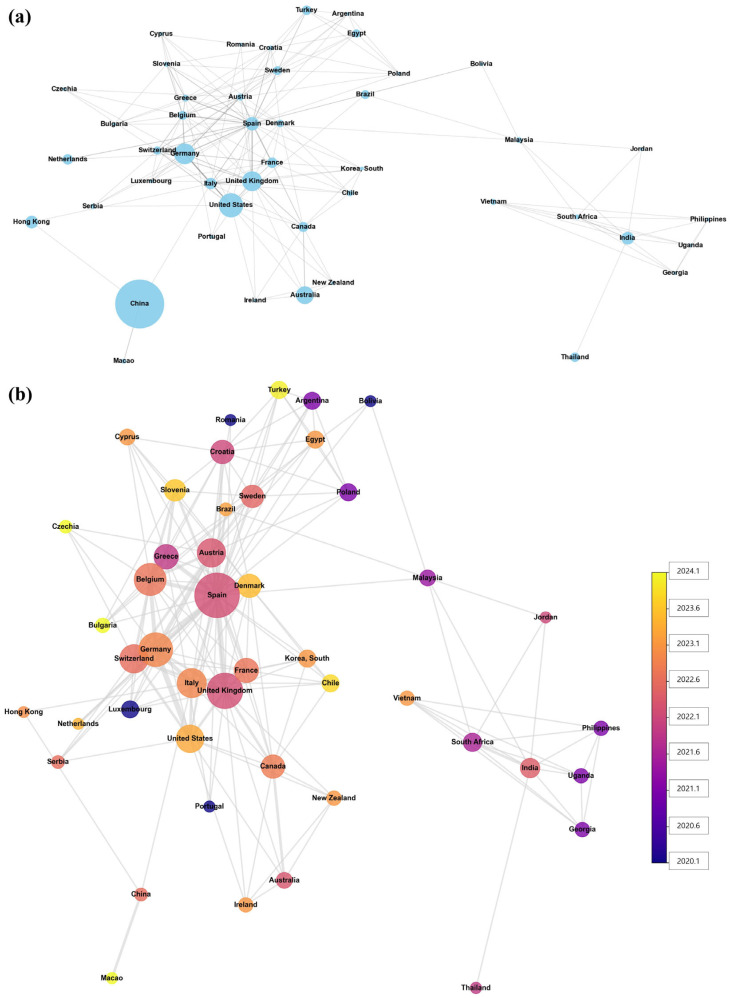
International collaboration networks in AI clinical trials. Note. This figure presents two network visualizations of international collaboration in AI clinical trials. (**a**) displays the network where node size indicates the total number of studies participated in (Full Count). (**b**) shows the network where node size represents the total collaboration connections (network centrality), and node color indicates the average research year (blue—earlier years, e.g., 2020.1; yellow—recent years, e.g., 2024.1). In both panels, lines (edges) denote collaborative relationships.

**Table 1 healthcare-13-02018-t001:** Hyperparameters for BERTopic topic modeling.

Algorithm Component	Parameter	Value
Embedding Model	Model	pritamdeka/S-BioBert-snli-multinli-stsb
UMAP	n_neighbors	5
	n_components	5
	min_dist	0.0
	metric	cosine
HDBSCAN	min_cluster_size	5
	min_samples	3
	metric	Euclidean
CountVectorizer	min_df	2
	max_features	1000
	ngram_range	(1, 2)
Topic Number	Determination	Automatic (data-driven)

Note. Topic classifications underwent manual review for clinical meaningfulness after automatic clustering. Abbreviations. HDBSCAN—Hierarchical Density-Based Spatial Clustering of Applications with Noise; UMAP—Uniform Manifold Approximation and Projection.

**Table 2 healthcare-13-02018-t002:** Country-level participation and collaboration in AI clinical trials.

Rank	Country/Region	Total Participation	Lead Studies	Multicountry Participation (%)	Single Country Studies	Total Partners	Avg. Partners per Study
1	China	238	238	3 (1.3)	235	4	1
2	United States	57	53	6 (10.5)	51	18	1.3
3	Germany	42	36	9 (21.4)	33	27	1.6
4	UK	38	31	7 (18.4)	31	30	1.8
5	Australia	30	30	2 (6.7)	28	6	1.2
6	Taiwan	19	19	0 (0)	19	0	1
7	Spain	17	6	11 (64.7)	6	47	3.8
8	Hong Kong	15	14	1 (6.7)	14	2	1.1
9	India	15	14	3 (20)	12	9	1.6
10	South Korea	14	14	0 (0)	14	0	1
11	Italy	12	7	7 (58.3)	5	20	2.7
12	Japan	11	11	0 (0)	11	0	1
13	Netherlands	10	9	2 (20)	8	3	1.3
14	France	10	8	2 (20)	8	14	2.4
15	Canada	9	7	3 (33.3)	6	13	2.4
16	Thailand	8	7	1 (12.5)	7	1	1.1
17	Turkey	8	7	1 (12.5)	7	7	1.9
18	Brazil	7	5	2 (28.6)	5	4	1.6
19	Belgium	7	5	5 (71.4)	2	24	4.4
20	Singapore	7	7	0 (0)	7	0	1

Note. This table provides a detailed overview of country-level participation in AI-related clinical trials, encompassing metrics such as the total number of trials a country participated in (Total Participation), the count of trials primarily led by the country (Lead Studies), and the distribution between multicountry participation and single-country studies. It also quantifies international engagement through the total number of unique partner countries (Total Partners), and the average number of partners per multicountry study (Avg. Partners Per Study).

## Data Availability

The datasets used and analyzed during the current study are available from the corresponding author on reasonable request.

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
