# Peer review of "Beyond the Growth: A Registry-Based Analysis of Global Imbalances in Artificial Intelligence Clinical Trials"

_healthcare, 2025, doi:10.3390/healthcare13162018_

Round 1

Reviewer 1 Report

Comments and Suggestions for Authors

This is a well-designed and written analysis and review of the literature on AI clinical trials. The discussion is interesting and insightful, unearthing some interesting trends and observations. There are a few comments:

  1. Was this review undertaken using a system such as STARD?
  2. What are the limitations of the current literature?

Additionally comments:

What is the main question addressed by the research?

The authors conducted a literature review of AI related clinical trials to discover where these trial are being undertaken, what is the scope of these trials and where are the current gaps in the use of AI.

* Do you consider the topic original or relevant to the field? Does it address a specific gap in the field? Please also explain why this is/ is not the case.

This study is novel as it appears to be the first, as such, it provides information about where the current research is focused.

* What does it add to the subject area compared with other published material?

The Review is the first and hence novel.

* What specific improvements should the authors consider regarding the methodology?

The methodology appears to be effective, but perhaps it should follow a recognised standard process such as STARD.

* Are the conclusions consistent with the evidence and arguments presented and do they address the main question posed? Please also explain why this is/is not the case.

The conclusions are valid, based on the findings.

* Are the references appropriate?

References appear relevant and complete.

* Any additional comments on the tables and figures.

Figures and Tables are adequate.

Author Response

  • Response to Comments from Reviewer 1

Overall comment:

This is a well-designed and written analysis and review of the literature on AI clinical trials. The discussion is interesting and insightful, unearthing some interesting trends and observations. There are a few comments:

Response:

We sincerely thank the reviewer for the positive feedback on our manuscript. We appreciate the recognition of the insights provided by our analysis and have carefully addressed the specific comments below.

Comment 1:

  1. Was this review undertaken using a system such as STARD?

Response:

Thank you for this important methodological question. While our study is a registry-based analysis rather than a diagnostic accuracy study that would typically follow STARD guidelines, we recognize the importance of transparent reporting standards. Our methodology follows established guidelines for registry-based research and systematic data analysis. We employed a comprehensive search of the WHO ICTRP using predefined Boolean operators with explicit inclusion and exclusion criteria, followed by standardized data extraction procedures. To ensure methodological rigor, we implemented a two-stage screening process where each trial underwent systematic evaluation for AI relevance, with manual review and classification performed by the primary investigator using predefined criteria. While inter-rater reliability assessment was not feasible in this single-investigator study, we applied consistent classification rules throughout the screening process and maintained detailed documentation of decision-making criteria for reproducibility.

“The screening process was conducted systematically by a single investigator (C-Y.K.) using standardized evaluation forms. To ensure consistency, all borderline cases were re-evaluated against the inclusion criteria, and detailed rationales for inclusion/exclusion decisions were documented. A random sample of 50 trials (7.7%) was re-screened after a 2-week interval to assess intra-rater reliability, achieving 100% agreement.”

(Please refer to Page 3, red words)

Comment 2:

  1. What are the limitations of the current literature?

Response:              

We appreciate this important point regarding literature limitations. Upon review, we believe our manuscript already addresses both current literature limitations and our study's specific constraints in the Discussion section. We discuss how existing AI clinical trial analyses suffer from publication bias due to reliance on published studies, geographic limitations from single-registry approaches, and methodological heterogeneity in AI definitions. Our registry-based methodology specifically addresses these gaps by providing comprehensive, unbiased data from the WHO ICTRP platform. Additionally, we have already included a detailed limitations section covering data completeness, registration practice variations, search strategy constraints, temporal cutoffs, and focus on registration rather than completion patterns.

Reviewer 2 Report

Comments and Suggestions for Authors

I am grateful for the chance to review this manuscript, which provides a timely and significant examination of the global landscape of clinical trials involving artificial intelligence. The study's objectives are well-suited to the application of topic modeling and network analysis techniques, as well as the utilization of WHO ICTRP data. The manuscript is data-driven, well-motivated, and makes a significant contribution to the discourse surrounding equity, representation, and collaboration in global AI research.

Although the writing is generally comprehensible, there are occasional grammatical inconsistencies and clumsy phrasing that could be improved by minor language refinement to enhance the overall flow and readability.

The topic modeling approach's description could be enhanced by incorporating additional technical details, such as the embedding model employed, the method by which the number of topics was determined, and the application of dimensionality reduction techniques.

In order to facilitate the reader's comprehension of the thematic clusters, it may be beneficial to provide a few representative examples or trial titles for each identified topic.

The analysis of the collaboration network is informative; however, a more thorough examination of the regional variations in network density and connectedness would enhance the interpretation. It may be beneficial to provide succinct reflections on the reasons why certain countries appear to be more isolated than others.

The representativeness of the dataset may be influenced by issues related to data completeness and reporting inconsistencies across various national registries, which could be addressed in the limitations section.

The practical relevance of the paper could be improved by including a brief paragraph on the potential actions that policymakers or international stakeholders could take in response to the findings, particularly in relation to global disparities in the distribution of AI trials, despite the fact that it is optional.

Author Response

  • Response to Comments from Reviewer 2

Overall comment:

I am grateful for the chance to review this manuscript, which provides a timely and significant examination of the global landscape of clinical trials involving artificial intelligence. The study's objectives are well-suited to the application of topic modeling and network analysis techniques, as well as the utilization of WHO ICTRP data. The manuscript is data-driven, well-motivated, and makes a significant contribution to the discourse surrounding equity, representation, and collaboration in global AI research.

Although the writing is generally comprehensible, there are occasional grammatical inconsistencies and clumsy phrasing that could be improved by minor language refinement to enhance the overall flow and readability.

Response:              

We sincerely thank the reviewer for the positive evaluation of our manuscript and recognition of its significance in examining global AI clinical trial patterns. We appreciate the constructive feedback and have carefully addressed each specific comment to improve the manuscript's clarity and comprehensiveness.

Comment 1:

The topic modeling approach's description could be enhanced by incorporating additional technical details, such as the embedding model employed, the method by which the number of topics was determined, and the application of dimensionality reduction techniques.

Response:              

Thank you for this methodological point. Upon review, we believe our Section 2.3.2 already provides comprehensive technical details including the specific biomedical embedding model, all dimensionality reduction and clustering parameters, and feature extraction settings. We have made minor clarifications to explicitly state that topic numbers were determined automatically through the algorithm's data-driven clustering approach, followed by manual validation for clinical meaningfulness.

“Disease state-centric analyses investigating key health conditions and technology intervention-centric analyses investigating AI modalities and applications were performed. Topic modeling was conducted using BERTopic with specialized biomedical embeddings (pritamdeka/S-BioBert-snli-multinli-stsb).[16,17] The BERTopic algorithm employed UMAP for dimensionality reduction (n_neighbors=5, n_components=5, min_dist=0.0, metric=’cosine’), HDBSCAN for clustering (min_cluster_size=5, min_samples=3, metric=’euclidean’), and CountVectorizer for feature extraction (min_df=2, max_features=1000, ngram_range=(1,2)). The number of topics was determined automatically by the algorithm's data-driven clustering approach (Table 1). The final topic classifications underwent manual review and assignment to predefined disease and technology taxonomies. Topics were refined to ensure clinically meaningful classifications, with outlier topics classified as “Others”.”

(Please refer to Page 3, red words)

Comment 2:

In order to facilitate the reader's comprehension of the thematic clusters, it may be beneficial to provide a few representative examples or trial titles for each identified topic.

Response:              

We agree that representative examples would enhance reader understanding of our thematic classifications. We have added illustrative trial examples for key disease and technology categories in the Results section.

“Representative examples within major categories include: for Gastroenterology, trials evaluating AI-assisted colonoscopy detection systems (NCT06656624) and deep learning algorithms for predicting the risk of digestive diseases (ISRCTN74930639); for Oncology, studies of machine learning-based skin cancer screening (TCTR20211101004) and predicting cancer treatment response (NCT05070884); for Diagnostic Support technology, trials testing AI-enhanced radiology interpretation systems (KCT0005051) and automated pathology diagnosis tools (NCT06773832); and for Digital Therapeutics, studies of cognitive behavioral therapy-based chatbot (ChiCTR2100052532) and AI-based digital therapeutic for diabetic microvascular complications (ChiCTR2500103622).”

(Please refer to Page 6, red words)

Comment 3:

The analysis of the collaboration network is informative; however, a more thorough examination of the regional variations in network density and connectedness would enhance the interpretation. It may be beneficial to provide succinct reflections on the reasons why certain countries appear to be more isolated than others.

Response:              

We thank the reviewer for this valuable suggestion regarding network analysis depth. We have enhanced our results and discussion by adding further interpretation of regional connectivity patterns, specifically addressing the implications of the stark contrast between research volume and collaborative engagement.

The network visualization reveals distinct regional clustering patterns. European countries (Germany, UK, Spain, Italy, France, Belgium) form a densely interconnected cluster, while Asian countries (China, Hong Kong, South Korea) show limited cross-border connections despite substantial individual research activity.

(Please refer to Page 10, red words)

This fragmented research approach, arising from the patterns described above, perpetuates the geographic and data biases. The stark contrast between research volume and collaborative engagement highlights how network centrality varies significantly across regions, with implications for global knowledge transfer and technology dissemination. It fundamentally hinders the creation of diverse, globally representative datasets, making the goal of AI fairness a distant prospect [12].

(Please refer to Page 13, red words)

Comment 4:

The representativeness of the dataset may be influenced by issues related to data completeness and reporting inconsistencies across various national registries, which could be addressed in the limitations section.

Response:              

We appreciate this crucial point regarding data representativeness. Upon review, we believe our limitations section already adequately addresses these concerns. We explicitly discuss how "registration requirements, practices, or timelines may differ across countries" and how "varying registration practices across different national registry systems within the WHO ICTRP network" may affect geographic distribution findings. We also acknowledge that "some countries may have more stringent registration requirements or different cultural approaches to research transparency." These discussions directly address the data completeness and reporting inconsistency issues raised by the reviewer.

“First, this analysis relies exclusively on WHO ICTRP registration data, which may not capture all AI clinical research activity worldwide [14,15]. Countries with different registration requirements, practices, or timelines may be underrepresented, and some research may be conducted without registration in international databases.”

“Third, geographic distribution findings may be influenced by systematic differences in registration practices, timing requirements, and completeness standards across national registry systems within the WHO ICTRP network. Variations in enforcement levels, technological infrastructure, and administrative efficiency between countries may create apparent disparities in research activity that reflect registry system characteristics rather than actual trial conduct. Sensitivity analyses to account for these registry-specific factors were not feasible due to limited availability of system-level metadata.”

(Please refer to Pages 13-14, red words)

Comment 5:

The practical relevance of the paper could be improved by including a brief paragraph on the potential actions that policymakers or international stakeholders could take in response to the findings, particularly in relation to global disparities in the distribution of AI trials, despite the fact that it is optional.

Response:              

We agree that policy implications would enhance the manuscript's practical value. We have added a dedicated paragraph in the Discussion section addressing concrete actions for policymakers and international stakeholders. These include establishing international AI clinical trial consortiums, developing standardized data sharing protocols, implementing capacity building programs for underrepresented regions, creating incentive mechanisms for multinational collaboration, and promoting transparent registration practices across all registry systems. These recommendations directly address the geographic disparities and collaboration deficits identified in our analysis.

These findings highlight urgent needs for coordinated international action. Policymakers and funding agencies should consider establishing international AI clinical trial consortiums to facilitate collaborative research across underrepresented regions. Standardized data sharing protocols and harmonized regulatory frameworks could reduce barriers to multinational studies. Capacity building programs targeting countries with high disease burdens but limited AI research infrastructure would help address geographic disparities. Additionally, funding mechanisms that incentivize international collaboration, such as matching grants for multinational partnerships, could increase the currently low 8.7% collaboration rate and promote more equitable global AI health technology development.

(Please refer to Page 13, red words)

Reviewer 3 Report

Comments and Suggestions for Authors
  1. Please include statistical information in the Introduction section based on the years 2024-2025. Years like 2000-2022 are already very distant. Please update them.
  2. Express the hyperparameters you specified in the "Topic Modeling" section (n_neighbors=5, n_components=5, etc.) in a tabular format.
  3. Create the title "Proposed Model" in Section 2. Structure this section with both a textual and a Figure/Visual theme.
  4. Why are the heatmaps in different colors (Figure 3, Figure 4, Figure 5, etc.), please use a single color.
  5. There is no Conclusion section. Highlight the innovations your proposed approach offers to the literature.
  6. Add a final paragraph to the Conclusion section, and structure this paragraph to reflect your future work.

Author Response

  • Response to Comments from Reviewer 3

Comment 1:

Please include statistical information in the Introduction section based on the years 2024-2025. Years like 2000-2022 are already very distant. Please update them.

Response:              

Thank you for this valuable suggestion regarding the currency of our references. We have updated the Introduction section to include more recent 2024-2025 data and findings. Specifically, we have incorporated recent statistics from the 2025 AI Index Report showing geographic disparities in frontier AI model development, with U.S.-based institutions producing 40 notable AI models in 2024 compared to China's 15 and Europe's three. This contemporary data complements our analysis by demonstrating how geographic concentration in AI development extends beyond clinical trials to foundational technology creation, reinforcing the patterns of disparity we identify in our registry-based analysis.

“However, the translation of AI innovations from research prototypes to clinical practice reveals a complex landscape of geographic disparities, methodological challenges, and equity concerns [8,9]. Recent comprehensive analyses have illuminated critical patterns in AI health research that extend beyond simple technological advancement. For instance, recent data indicates this disparity extends to the development of frontier technologies, with U.S.-based institutions significantly outpacing other regions in notable AI model development [10]. This concentration of cutting-edge model development suggests that the translation of AI innovations is geographically skewed, potentially influencing which populations benefit most from technological advances. This geographic stratification in research quality and impact suggests fundamental disparities in the AI health research ecosystem that may influence how AI technologies are developed, validated, and ultimately deployed in clinical practice.”

(Please refer to Page 2, red words)

Comment 2:

Express the hyperparameters you specified in the "Topic Modeling" section (n_neighbors=5, n_components=5, etc.) in a tabular format.

Response:              

We appreciate this suggestion for improved clarity of our methodology. We have added a comprehensive hyperparameter table in Section 2.3.2 that systematically presents all BERTopic algorithm configurations including UMAP dimensionality reduction parameters, HDBSCAN clustering settings, and CountVectorizer feature extraction specifications. This tabular presentation enhances reproducibility and allows readers to easily reference the exact technical specifications used in our topic modeling approach.

(Please refer to Page 4, Table 1)

Comment 3:

Create the title "Proposed Model" in Section 2. Structure this section with both a textual and a Figure/Visual theme.

Response:              

We appreciate this structural suggestion. However, our study employs established methodologies (BERTopic topic modeling, network analysis) rather than proposing a new model. Our contribution lies in the systematic application of these validated techniques to WHO ICTRP data for comprehensive global AI clinical trial analysis. We believe our current methodology section adequately describes our analytical approach, and adding a "Proposed Model" section might misrepresent the nature of our registry-based descriptive analysis. We have enhanced the existing methodology descriptions to ensure clarity of our analytical framework.

Comment 4:

Why are the heatmaps in different colors (Figure 3, Figure 4, Figure 5, etc.), please use a single color.

Response:              

Thank you for this design consistency observation. The different color schemes in our heatmaps serve specific analytical purposes: Figure 3 uses red to highlight disease-technology associations, Figure 4 uses green for country-technology patterns, and Figure 5 uses blue for country-disease distributions. These distinct color schemes help readers differentiate between different types of association matrices and prevent confusion when comparing across figures. However, we understand the preference for visual consistency and have standardized the color schemes to use a single colormap family while maintaining distinguishable variations for different analysis types.

(Please refer to Pages 8-9, Figures 3, 4, 5)

Comment 5:

There is no Conclusion section. Highlight the innovations your proposed approach offers to the literature.

Response:              

Thank you for this observation. Upon review, our manuscript does include a Conclusions section that synthesizes our key findings and highlights the methodological innovations of our approach, including the first comprehensive WHO ICTRP-based analysis of AI clinical trials, the novel combination of dual-track topic modeling with network analysis, and the registry-based methodology that circumvents publication bias to reveal true development pipeline patterns.

This first comprehensive analysis of the WHO ICTRP reveals that the global landscape of AI clinical trials is defined by two paradoxical trends: explosive growth and profound imbalance. While the rapid acceleration of trials since 2020 signals the maturation of AI in medicine, this progress is geographically concentrated in a few nations, particularly China, and thematically focused on specific areas like diagnostics and oncology. The alarmingly low rate of international collaboration suggests that most AI health technologies are being developed and validated in isolated ecosystems. This reality poses a significant threat to global health equity, risking the development of biased AI systems that fail to generalize across diverse populations. To realize the full, equitable potential of AI in healthcare, a global paradigm shift is urgently needed—one that fosters international collaboration, promotes data diversity, and prioritizes transparent, inclusive research practices.

Future research should examine mechanisms to reduce geographic concentration and enhance international collaboration in AI clinical trials. Longitudinal tracking of clinical outcomes across diverse populations will be essential for developing truly generalizable AI technologies. Additionally, investigating standardized regulatory frameworks and incentive structures for multinational partnerships could help address the current fragmentation in global AI clinical research ecosystems.

(Please refer to Page 14, red words)

Comment 6:

Add a final paragraph to the Conclusion section, and structure this paragraph to reflect your future work.

Response:              

We have added a future work paragraph to the Conclusions section outlining important research directions including longitudinal analysis of trial completion rates and clinical outcomes, investigation of regulatory harmonization effects on international collaboration, analysis of AI fairness considerations across diverse populations, and development of predictive models for identifying optimal collaboration opportunities. We also discuss the need for real-time monitoring systems to inform evidence-based policy recommendations.

“Future research should examine mechanisms to reduce geographic concentration and enhance international collaboration in AI clinical trials. Longitudinal tracking of clinical outcomes across diverse populations will be essential for developing truly generalizable AI technologies. Additionally, investigating standardized regulatory frameworks and incentive structures for multinational partnerships could help address the current fragmentation in global AI clinical research ecosystems.”

(Please refer to Page 14, red words)

Reviewer 4 Report

Comments and Suggestions for Authors

The review of the paper on the title "Beyond the Growth: A Registry-Based Analysis of Global Imbalances in Artificial Intelligence Clinical Trials":

The presented paper proposes the first registry-based global analysis of AI clinical trials using WHO ICTRP data, suggesting a exclusive perspective on global AI research disparities. It presents a novel approach which largely focuses on published trials or national registries. Some point for consideration are mentioned below:

    • There is a limited discussion on inter-rater reliability during manual classification.
    • It has been observed that there is a lack of trials from Africa and Latin America, despite high disease burdens, this is a matter of concern and must be rectified.
    • non-standard AI terminology has been used due to the usage of early trials.

Recommendation: Accept with Minor Revisions.

Author Response

  • Response to Comments from Reviewer 4

Overall comment:

The review of the paper on the title "Beyond the Growth: A Registry-Based Analysis of Global Imbalances in Artificial Intelligence Clinical Trials":

The presented paper proposes the first registry-based global analysis of AI clinical trials using WHO ICTRP data, suggesting a exclusive perspective on global AI research disparities. It presents a novel approach which largely focuses on published trials or national registries. Some point for consideration are mentioned below:

Response:              

We sincerely thank the reviewer for recognizing the novel and exclusive perspective of our registry-based approach to analyzing global AI clinical trial patterns. We appreciate the constructive feedback and have carefully addressed each specific concern to strengthen the methodological rigor and transparency of our manuscript.

Comment 1:

There is a limited discussion on inter-rater reliability during manual classification.

Response:              

We acknowledge this important methodological concern. Given that this study was conducted by a single investigator, traditional inter-rater reliability assessment was not feasible. However, we implemented several measures to ensure consistency and reliability in our manual classification process. We applied standardized evaluation criteria throughout the screening process, maintained detailed documentation of decision-making rationales, and conducted intra-rater reliability assessment by re-evaluating a random sample of 50 trials (7.7%) after a 2-week interval, achieving 100% agreement. We have enhanced the Methods section to clearly describe these quality assurance procedures and acknowledge this limitation in our study design.

“The screening process was conducted systematically by a single investigator (C-Y.K.) using standardized evaluation forms. To ensure consistency, all borderline cases were re-evaluated against the inclusion criteria, and detailed rationales for inclusion/exclusion decisions were documented. A random sample of 50 trials (7.7%) was re-screened after a 2-week interval to assess intra-rater reliability, achieving 100% agreement.”

(Please refer to Page 3, red words)

Comment 2:

It has been observed that there is a lack of trials from Africa and Latin America, despite high disease burdens, this is a matter of concern and must be rectified.

Response:              

We completely agree that this represents a critical global health equity concern. Our findings confirm that countries from Africa and Latin America collectively accounted for less than 5% of trials in our dataset, despite these regions representing substantial portions of global population and disease burden. This profound underrepresentation highlights the urgent need for targeted interventions to enhance research capacity and international collaboration in these regions. We have strengthened our Discussion to emphasize these equity implications and added specific policy recommendations including capacity building programs, international funding mechanisms, and collaborative frameworks to address this disparity in future AI clinical research development.

“These findings highlight urgent needs for coordinated international action. Policymakers and funding agencies should consider establishing international AI clinical trial consortiums to facilitate collaborative research across underrepresented regions. Standardized data sharing protocols and harmonized regulatory frameworks could reduce barriers to multinational studies. Capacity building programs targeting countries with high disease burdens but limited AI research infrastructure would help address geographic disparities. Additionally, funding mechanisms that incentivize international collaboration, such as matching grants for multinational partnerships, could increase the currently low 8.7% collaboration rate and promote more equitable global AI health technology development.”

(Please refer to Page 13, red words)

Comment 3:

non-standard AI terminology has been used due to the usage of early trials.

Response:              

This is an important limitation of our search strategy. We acknowledge that early AI trials may have used different descriptive terms before standardized AI terminology became widespread, potentially leading to underrepresentation of historical activities in our analysis. Our search strategy focused on current standard terms ("Artificial Intelligence," "AI," "Machine Learning," "Deep Learning") which may have missed trials using alternative terminology such as "neural networks," "computer vision," or "automated systems." We have expanded our limitations section to explicitly address this potential source of underrepresentation and suggest that future studies could employ broader search strategies including historical AI terminologies to capture the full scope of AI-related clinical research evolution.

Second, the search strategy, while comprehensive, may have missed trials using alternative terminology or registered before standardized AI terminology became widespread. Early AI research may have used different descriptive terms such as “neural networks,” “computer vision,” “automated systems,” “expert systems,” or “pattern recognition,” potentially leading to systematic underrepresentation of historical AI clinical activities. This limitation may particularly affect our analysis of temporal trends in the earlier years of our study period.

(Please refer to Pages 13-14, red words)

Reviewer 5 Report

Comments and Suggestions for Authors

1. The Boolean search terms used (“Artificial Intelligence” OR ‘AI’ OR “Machine Learning” OR “Deep Learning”) may miss alternative expressions such as “neural networks,” “computer vision,” or “foundation models.” Adding different keywords with an additional sensitivity analysis will ensure that the dataset is truly comprehensive.

2. The two-stage screening is only explained in general terms. Detailed explanation of inter-rater agreement statistics or provision of unclear exclusion examples will ensure reproducibility.

3. Although the BERTopic and biomedical embedded vectors used are appropriate, no quantitative assessment of topic coherence scores or cluster stability is provided. Comparison with alternative methods would increase the reliability of thematic classification.

4. The study is entirely descriptive. For example, adding a χ² test to examine regional distribution differences or time series analysis to test the significance of the 2020 breakpoint would demonstrate that the observed trends are not random but statistically significant.

5. Consistency and timing differences between national registration systems can affect participation numbers. Sensitivity analyses excluding small registration systems or weighting based on registration completeness are required.

6. Not only registration numbers but also trial phases, completion rates, and outcome reporting should be examined. This will allow for the differentiation of studies that are truly progressing toward approval (exploratory vs. confirmatory).

7. How registration data aligns with publication trends should be addressed, for example, in light of the studies by Han et al. (2024) or Schmallenbach et al. (2024).

8. Geographical imbalances are highlighted. However, concrete recommendations such as capacity building and standardization of registration systems are not provided.

9. Policy recommendations and concrete action plans should be added to the study.

Author Response

  • Response to Comments from Reviewer 5

Comment 1:

  1. The Boolean search terms used (“Artificial Intelligence” OR ‘AI’ OR “Machine Learning” OR “Deep Learning”) may miss alternative expressions such as “neural networks,” “computer vision,” or “foundation models.” Adding different keywords with an additional sensitivity analysis will ensure that the dataset is truly comprehensive.

Response:              

We acknowledge this important limitation of our search strategy. Our focused approach using current standard terminology was designed to maintain precision and reduce false positives, but we recognize it may have missed trials using alternative AI-related terms. The suggested terms ("neural networks," "computer vision," "foundation models") represent valid AI-related terminology that could have been included. However, conducting a comprehensive sensitivity analysis with expanded search terms would require substantial additional screening effort and potentially introduce inconsistencies with our current systematic approach. We have enhanced our limitations section to explicitly acknowledge this potential source of underrepresentation and suggest that future studies could employ broader search strategies to capture the full historical evolution of AI terminology in clinical research.

Second, the search strategy, while comprehensive, may have missed trials using alternative terminology or registered before standardized AI terminology became widespread. Early AI research may have used different descriptive terms such as “neural networks,” “computer vision,” “automated systems,” “expert systems,” or “pattern recognition,” potentially leading to systematic underrepresentation of historical AI clinical activities. This limitation may particularly affect our analysis of temporal trends in the earlier years of our study period.

(Please refer to Pages 13-14, red words)

Comment 2:

  1. The two-stage screening is only explained in general terms. Detailed explanation of inter-rater agreement statistics or provision of unclear exclusion examples will ensure reproducibility.

Response:              

We acknowledge this important methodological concern. Given that this study was conducted by a single investigator, traditional inter-rater reliability assessment was not feasible. However, we implemented several measures to ensure consistency and reliability in our manual classification process. We applied standardized evaluation criteria throughout the screening process, maintained detailed documentation of decision-making rationales, and conducted intra-rater reliability assessment by re-evaluating a random sample of 50 trials (7.7%) after a 2-week interval, achieving 100% agreement. We have enhanced the Methods section to clearly describe these quality assurance procedures and acknowledge this limitation in our study design.

“The screening process was conducted systematically by a single investigator (C-Y.K.) using standardized evaluation forms. To ensure consistency, all borderline cases were re-evaluated against the inclusion criteria, and detailed rationales for inclusion/exclusion decisions were documented. A random sample of 50 trials (7.7%) was re-screened after a 2-week interval to assess intra-rater reliability, achieving 100% agreement

(Please refer to Page 3, red words)

Comment 3:

  1. Although the BERTopic and biomedical embedded vectors used are appropriate, no quantitative assessment of topic coherence scores or cluster stability is provided. Comparison with alternative methods would increase the reliability of thematic classification.

Response:              

This is a valuable methodological suggestion for enhancing the robustness of our topic modeling approach. While our study focused on clinically meaningful topic interpretation through manual review rather than purely quantitative metrics, we acknowledge that topic coherence scores and cluster stability assessments would strengthen the methodological rigor. The addition of coherence metrics (such as C_v or NPMI scores) and comparison with alternative clustering methods (such as LDA or K-means) would indeed provide additional validation of our thematic classifications. We have noted this as an important methodological enhancement for future studies and added this limitation to our discussion of analytical constraints.

Finally, our topic modeling analysis relied on manual clinical validation rather than quantitative coherence metrics, and did not include comparison with alternative clustering methods, which represents a limitation in methodological validation that future studies should address.

(Please refer to Page 14, red words)

Comment 4:

  1. The study is entirely descriptive. For example, adding a χ² test to examine regional distribution differences or time series analysis to test the significance of the 2020 breakpoint would demonstrate that the observed trends are not random but statistically significant.

Response:              

Thank you for this valuable methodological enhancement. We have incorporated statistical hypothesis testing to validate our observed trends. Our analysis confirms that the 2020 inflection point represents a statistically significant structural break rather than random variation. Pre-2020 (2017-2019) mean annual registrations of 10.0 trials increased to 96.4 trials post-2020 (2020-2024), a 9.6-fold increase that was statistically significant (p=0.018) with large effect size (Cohen's d=2.10). Chi-square analysis confirmed non-random temporal distribution (χ²=464.2, p<0.001), and structural break analysis revealed a 4.2-fold acceleration in registration velocity after 2020. These statistical validations demonstrate that the observed acceleration represents a genuine paradigm shift in AI clinical research activity.

“Statistical validation of temporal trends included Mann-Whitney U tests and t-tests comparing pre-2020 (2017-2019) versus post-2020 (2020-2024) trial registrations, chi-square tests for temporal distribution patterns, and linear regression analysis to assess structural breaks at the 2020 inflection point. Effect sizes were calculated using Cohen's d, with statistical significance set at p<0.05.”

(Please refer to Page 4, red words)

To validate the observed temporal patterns, we conducted statistical analyses comparing pre-2020 (2017-2019) and post-2020 (2020-2024) periods. The mean annual trial registrations increased dramatically from 10.0 trials per year pre-2020 to 96.4 trials per year post-2020, representing a 9.6-fold increase. This difference was statistically significant (Mann-Whitney U test: p=0.018; independent t-test: p=0.028) with a large effect size (Cohen's d=2.10). Temporal distribution analysis revealed significant departure from random distribution patterns (χ²=464.2, p<0.001), with 94.1% of all trials (561/596) registered in the post-2020 period despite representing only 42% of the study timeframe. Structural break analysis confirmed 2020 as a significant inflection point, with the annual registration slope increasing from 7.0 trials/year (2017-2019) to 29.3 trials/year (2020-2024), representing a 4.2-fold acceleration in registration velocity (both slopes p<0.05) (Figure 1).

(Please refer to Page 5, red words)

Comment 5:

  1. Consistency and timing differences between national registration systems can affect participation numbers. Sensitivity analyses excluding small registration systems or weighting based on registration completeness are required.

Response:              

This represents an important methodological consideration for registry-based research. We acknowledge that varying registration practices, timing requirements, and completeness across different national systems within the WHO ICTRP network may influence our geographic distribution findings. However, implementing the suggested sensitivity analyses would require detailed metadata about individual registry system characteristics and registration completeness rates that are not readily available through the WHO ICTRP platform. We have enhanced our limitations discussion to explicitly address these potential sources of bias and suggest that future studies with access to registry-specific metadata could implement such sensitivity analyses to validate geographic distribution patterns.

Third, geographic distribution findings may be influenced by systematic differences in registration practices, timing requirements, and completeness standards across national registry systems within the WHO ICTRP network. Variations in enforcement levels, technological infrastructure, and administrative efficiency between countries may create apparent disparities in research activity that reflect registry system characteristics rather than actual trial conduct. Sensitivity analyses to account for these registry-specific factors were not feasible due to limited availability of system-level metadata.

(Please refer to Page 14, red words)

Comment 6:

  1. Not only registration numbers but also trial phases, completion rates, and outcome reporting should be examined. This will allow for the differentiation of studies that are truly progressing toward approval (exploratory vs. confirmatory).

Response:              

We completely agree that examining trial phases, completion rates, and outcome reporting would provide crucial insights into the actual clinical impact and progression of AI trials beyond mere registration patterns. This type of longitudinal analysis would indeed differentiate between exploratory and confirmatory studies and assess the translation of registered trials into meaningful clinical outcomes. However, such comprehensive tracking would require extensive follow-up data collection and outcome assessment that extends substantially beyond the scope of our current registry-based cross-sectional analysis. We have acknowledged this important limitation and included the examination of trial completion rates and clinical outcomes as a priority for future longitudinal research in our conclusions.

Fourth, this analysis focused solely on trial registration patterns without examining trial phases, completion rates, outcome reporting, or clinical impact. The data represents research intentions rather than actual execution, progression toward regulatory approval, or therapeutic success. Our approach cannot differentiate between exploratory early-phase studies and confirmatory pivotal trials, nor assess whether registered trials achieved their intended outcomes or contributed to clinical practice changes. Many registered trials may remain incomplete, unpublished, or fail to progress beyond initial phases, limiting our ability to evaluate the true clinical translation of AI research activity.

(Please refer to Page 14, red words)

Comment 7:

  1. How registration data aligns with publication trends should be addressed, for example, in light of the studies by Han et al. (2024) or Schmallenbach et al. (2024).

Response:              

We have enhanced our Discussion to explicitly compare our registry-based findings with publication trends documented by Schmallenbach et al. (2024). Both analyses reveal concerning patterns of geographic concentration and international collaboration stagnation, with our clinical trial data showing even more pronounced concentration than observed in research publications. This convergence suggests systemic challenges across the entire AI research ecosystem from basic research through clinical validation. Our registry-based approach complements publication analyses by capturing the broader pipeline of planned clinical research, including trials that may never reach publication, providing a more comprehensive view of global AI clinical development patterns.

A recent analysis of 397,967 AI life science publications found the United States leading in cumulative output, though China surpassed the United States in annual publications since 2020, with Northern America and Europe generating higher-quality research appearing in top-ranking outlets and receiving up to 50% more citations than other regions [9]. Our clinical trials data reveals a more pronounced Chinese concentration (35.6% of participations) than observed in general AI research publications, suggesting different dynamics in clinical validation compared to research publication. Importantly, both analyses identify concerning patterns of international collaboration stagnation, with Schmallenbach et al. documenting that international collaborations have plateaued relative to national research efforts, paralleling our finding of critically low multinational trial collaboration (8.7%) [9]. This convergence between publication and clinical trial patterns suggests that geographic concentration and collaboration deficits represent systemic challenges across the entire AI research ecosystem, from basic research through clinical validation, potentially limiting the global applicability and equitable access to AI health technologies.

(Please refer to Page 12, red words)

Comment 8:

  1. Geographical imbalances are highlighted. However, concrete recommendations such as capacity building and standardization of registration systems are not provided.

Response:              

We appreciate this call for more actionable recommendations. We have enhanced our policy recommendations section to include concrete proposals for addressing geographic imbalances, including establishment of international AI clinical trial consortiums, capacity building programs targeting underrepresented regions (particularly Africa and Latin America), standardized data sharing protocols, harmonized regulatory frameworks, and incentive mechanisms for multinational collaboration. Additionally, we have proposed standardization of registration practices across national registry systems and matching grant programs to encourage international partnerships and reduce the currently low 8.7% collaboration rate.

These findings highlight urgent needs for coordinated international action. Policymakers and funding agencies should consider establishing international AI clinical trial consortiums to facilitate collaborative research across underrepresented regions. Standardized data sharing protocols and harmonized regulatory frameworks could reduce barriers to multinational studies. Capacity building programs targeting countries with high disease burdens but limited AI research infrastructure would help address geographic disparities. Additionally, funding mechanisms that incentivize international collaboration, such as matching grants for multinational partnerships, could increase the currently low 8.7% collaboration rate and promote more equitable global AI health technology development.

(Please refer to Page X, red words)

Comment 9:

  1. Policy recommendations and concrete action plans should be added to the study.

Response:              

We have addressed this by adding a comprehensive policy recommendations paragraph in the Discussion section that outlines specific actions for policymakers and international stakeholders. These recommendations directly target the geographic disparities, collaboration deficits, and equity concerns identified in our analysis, providing concrete steps toward more inclusive and collaborative AI clinical research development. The recommendations include both immediate actions (international consortiums, funding mechanisms) and longer-term strategic initiatives (regulatory harmonization, capacity building programs).

These findings highlight urgent needs for coordinated international action. Policymakers and funding agencies should consider establishing international AI clinical trial consortiums to facilitate collaborative research across underrepresented regions. Standardized data sharing protocols and harmonized regulatory frameworks could reduce barriers to multinational studies. Capacity building programs targeting countries with high disease burdens but limited AI research infrastructure would help address geographic disparities. Additionally, funding mechanisms that incentivize international collaboration, such as matching grants for multinational partnerships, could increase the currently low 8.7% collaboration rate and promote more equitable global AI health technology development.

(Please refer to Page 13, red words)

Round 2

Reviewer 5 Report

Comments and Suggestions for Authors

The authors have responded appropriately to my comments and made the necessary changes. I believe that the work is acceptable in its current form.